# Nursing and Telemental Health during the COVID-19 Pandemic

**DOI:** 10.3390/healthcare10020273

**Published:** 2022-01-30

**Authors:** Antonio José Sánchez-Guarnido, María Gonzalez-Vilchez, Rosario de Haro, Magdalena Fernández-Guillen, Mireia Graell-Gabriel, Valentina Lucena-Jurado

**Affiliations:** 1Hospital Santa Ana, 18600 Motril, Spain; antonioj.sanchez.guarnido.sspa@juntadeandalucia.es (A.J.S.-G.); maria.gonzalez.vilchez.sspa@juntadeandalucia.es (M.G.-V.); rosario.haro.sspa@juntadeandalucia.es (R.d.H.); magdalena.fernandez.sspa@juntadeandalucia.es (M.F.-G.); 2Hospital Clínic de Barcelona, 08036 Barcelona, Spain; mgraell@clinic.cat; 3Facultad de Ciencias de la Educación, Universidad de Córdoba, 14071 Córdoba, Spain

**Keywords:** nursing, serious mental illness, telemental health

## Abstract

Measures taken to reduce the rate of contagion during the first months of the COVID-19 pandemic in Spain led to the interruption of nursing interventions for many patients with serious mental illness (SMI), while others stayed in touch with their nurses telematically. However, published research into the impact of mental telehealth and the outcome of the changes that took place in the pandemic is very limited. **Aim:** The aim of this study was to analyze the changes in nursing interventions received by severe mental illness (SMI) patients and to test whether telemental health (TH) has been effective in reducing relapses during the COVID-19 pandemic. **Materials and methods:** Information was gathered retrospectively from about 270 patients under treatment at 15 mental health day hospitals (MHDHs) in Spain during the year 2020. **Results:** Face-to-face nursing interventions were found to have decreased and TH interventions were found to have increased in the first few months of the pandemic. In the following months, TH interventions—especially those conducted by video call—helped reduce the number of relapses. **Conclusions:** TH helps provide news forms of effective telematic nursing interventions that reduce the number of relapses and admissions in patients with serious mental health disorders.

## 1. Introduction

The much-increased likelihood of an increase in the number and severity of mental health problems and the worsening of pre-existing psychiatric pathologies have led some authors to define their effects on mental health as a second pandemic [1]. Different studies have linked COVID-19 infection to the development of mental pathologies [2,3]. Lockdown, too, can have negative psychological effects [4,5]. A study in Spain found high levels of sleeplessness and emotional symptoms during the lockdown period [6].

One of the risk factors for developing mental problems related to COVID-19 is the presence of a previous mental pathology [7]. Greater increases in anxiety levels and a worsening of previous pathologies have been reported in populations already suffering from mental disorders [7,8]. Social isolation may also have a greater effect on these patients. Social support is, after all, associated with better recovery, hence the attempts to encourage broad community support and social integration [9].

All these reasons make it necessary to develop alternatives that will keep patients with mental illness in contact with the resources and treatments they need in a pandemic situation. However, government handling of the first wave of the pandemic limited the number of face-to-face interventions. On the one hand, this reduced the number of infections, but on the other, it led to many nurses being relocated to other services to attend to COVID-19 patients [10,11,12,13].

To alleviate the consequences of this disconnection as much as possible, telemental health (TH) services were implemented. These included the use of mobile devices, video calls, email, videos, and other web-based resources [14]. A mixed model combining face-to-face and telematic interventions was also recommended, including the personalized design of treatments [15]. In some health areas, face-to-face mental health consultations were reserved for the administration of intramuscular treatment and the evaluation of people with more severe illnesses, while other patients were attended by telephone [16].

Before the pandemic, professionals had expressed numerous concerns regarding the implementation of TH, citing the loss of therapeutic relationships, teamwork, confidentiality and privacy, the insecurity and the legal regulation of personal data transmission, the need for professional recycling and training [14,15,17,18], and the need for patients to have access to the material resources (mobile phones, internet connection) and skills necessary to access this care [15].

Due to limited research into the issue, no conclusive results have yet been obtained about the effectiveness of TH in mental health nursing [19]. However, some studies have shown that the use of TH may be helpful to people with mental disorders [20]. TH undoubtedly provides mental health nurses with opportunities to transform a traditional nursing practice in a context like that generated by the pandemic. Harnessing the power of technology may offer an innovative alternative means of providing quality mental health care which also makes that care more accessible [14], but more research is needed to test this premise. The objectives of this study were therefore to analyze the changes that occurred in mental health nursing interventions during the pandemic, to test whether telematic interventions were effective in reducing patient relapse, and to analyze the existence of possible differences in the effectiveness of the different telematic formats used.

## 2. Materials and Methods

### 2.1. Design and Participants

A retrospective cohort study was carried out. 15 mental health day hospitals (MHDH) were selected using sampling stratified by regions in order to facilitate generalization to the Spanish population. Based on the resulting estimate, 270 people were included in the study. All were over 18 years of age, diagnosed with serious mental illness (SMI), and had been under follow-up in MHDHs during 2020. The calculation of n through G*Power analysis brought the sample size closer to 272. Serious mental illness was defined not exclusively at the level of psychopathological diagnosis, but rather as a function of the severity and intensity of the required intervention. This criterion was implemented by selecting all patients who, due to the severity of their illness, required follow-up in one of the MHDHs participating in the study during this period.

Collaborators in each MHDH collected the data retrospectively from the patients’ clinical histories between October and November 2020. To guarantee capacity, coherence, and correctness in the collection of data from the 15 centers, three online training sessions were held for the professionals involved. A password-protected database was designed and equipped with different logical mechanisms that prevented the introduction of erroneous data. The database was anonymized and used exclusively by the researchers.

Data was collected about the main mental health diagnoses and also about sociodemographic variables such as age, sex, level of education., household composition, and employment status.

For the first objective, the time period in relation to the epidemiological situation and the state of lockdown in Spain was used as the exposure variable. Three observation periods of two months were established: the period prior to the pandemic (from 16 January to 15 March 2020), the lockdown period (from 16 March to 15 May 2020, and the de-escalation period (from 16 May to 15 July 2020).

Nursing interventions received by patients in person, by telephone, by videoconference, or by other telematic means (messaging, Facebook, e-mail, or blog) were used as response variables. They were coded dichotomously according to whether they had been received or not.

For the second objective, the independent variable used was having received some type of TH nursing care during lockdown. For this second objective, the primary response variable was the number of full hospital admissions during the following six months.

Finally, for the third objective, each of the three types of telematic channels employed (telephone, video call, and other telematic formats) was used as an independent variable and the percentage of admissions to full hospitalization at six months was used as a response variable. Telephone and videoconference interventions were carried out weekly with individual patients by professionals who already knew the patient, thus facilitating identification and good communication. Each telematic intervention had a duration of 15–30 min and was performed via telephone/corporate application to ensure the confidentiality of each interview.

### 2.2. Data Analysis

All analyses were performed using IBM-SPSS V.21.0 (IBM Corp., Armonk, NY, USA), with a statistical significance value of *p* < 0.05. Percentages were used for the categorical variable results and mean and standard deviation were used for the quantitative variables.

### 2.3. Bivariate Analysis

To examine the first objective, a series of McNemar tests was carried out comparing patients’ reception or non-reception of the different types of intervention in the three time periods (before, during, and after the first wave of the pandemic).

For the second objective, chi-square analyses were carried out to compare the proportion of hospitalizations of those patients who received some kind of telematic intervention during the lockdown period and those who did not.

Finally, for the third objective, chi-square analyses were carried out to compare the three types of telematic intervention received (telephone, videoconference, and other means) in relation to the percentage of admissions to MHDHs at six months after lockdown. In the cases in which it was not possible to use chi-square, Fisher’s exact test was used.

### 2.4. Multivariate Analysis

A multilevel logistic regression analysis was then performed, thus adjusting the results to possible confounding variables and possible interdependence effects with the interventions received before and after lockdown. Two levels were established: level 1 for the interventions received (before, during, and after lockdown) and level 2 for the characteristics of the subjects. A model was obtained with level 1 variables which included telephone interventions, videoconference interventions, interventions carried out by other telematic means, and person-to-person interventions, and level 2 variables which included both sociodemographic variables (sex and age) and clinical variables (diagnosis and adherence to treatment). The frequency of admissions at six months was kept as a dependent variable.

### 2.5. Ethical Considerations

The project was approved by the research ethics committee and the principles of the Helsinki Declaration were complied with at all times, as were the latest European Union (EU) regulations regarding data confidentiality. Each patient was informed about the project’s objectives and methodology and was asked to participate voluntarily.

## 3. Results

### 3.1. Descriptive Analysis of the Sample

As can be seen in Table 1, data were collected from 120 men and 150 women, aged between 18 and 67 years and with a mean age of 39.90 years (SD = 11.814). The sample was made up of people with psychotic disorders (30.4%), personality disorder (27.8%), bipolar disorder (10.4%), and severe depressive disorder (9.6%). 35.8% of the participants had a primary level of education, 41.5% had a secondary level of education, and 14.8% had attended university. 42.6% lived with a complete family of origin, 28.9% lived in their own family home, and 17% lived alone. With regard to their work situation, most of the participants were retired (29.3%) or unemployed (26.3%), and only 16.7% were working.

### 3.2. Analysis of Nursing Interventions Received Pre-Pandemic, during Lockdown, and Post-Pandemic

Table 2 shows the percentages of patients who received the different types of nursing interventions. The percentage receiving face-to-face interventions dropped from 59.6% before lockdown to only 16.3% during lockdown, before recovering to 60.4% in the subsequent period. The differences are significant between the pre-lockdown period and the lockdown period (χ^2^ = 104.310; *p* < 0.001) and between the lockdown period and the post-lockdown period (χ^2^ = 113.203; *p* < 0.001) but are not significant between pre-lockdown and post-lockdown (χ^2^ = 0.020; *p* = 0.888). The percentage of patients who received telephone calls rose from 4.1% before lockdown to 33.3% during lockdown, and then fell to 6.7% in the post-lockdown period. These differences are statistically significant between all the periods ((χ^2^ = 75.111; *p* < 0.001), (χ^2^ = 30.420; *p* < 0.001), and (χ^2^ = 20.338; *p* < 0.001)). The percentage of patients attended by videoconference rose from 0% before lockdown to 8.9% during lockdown and then fell to 5.2% afterwards. These differences are significant between the pre-lockdown period and both the lockdown period and the post-lockdown period (*p* < 0.001), but not between the lockdown and post-lockdown periods (*p* = 0.064). Other remote interventions were again almost nonexistent before lockdown (only one intervention), but were then received by 12.6% of patients during lockdown and by 9.3% after lockdown. These differences are statistically significant between the pre-lockdown period and both the lockdown period (χ^2^ = 29.257; *p* < 0.001) and the post-lockdown period (*p* < 0.001), but were not so significant between lockdown and post-lockdown (χ^2^ = 2.370; *p* = 0.124).

### 3.3. Analysis of Relapse as a Function of Telematic Nursing Intervention

Table 1 shows the characteristics of the two comparison groups. No significant differences can be seen in diagnosis (χ^2^ = 6.502; *p* = 0.165), age (t = 0.014; *p* = 0.989), sex (χ^2^ = 0.019; *p* = 0.889), or level of education (χ^2^ = 5.785; *p* = 0.216). There are, however, differences in the composition of the households in which the participants lived, with the intervention group having fewer people living alone (9.3% vs. 22.1%) and more people living in their own family homes (35.5% vs. 24.5%). There are also differences in work activity, with only 6.6% working in the intervention group as opposed to 23.3% in the control group (χ^2^ = 19.637; *p* < 0,01).

We next analyzed hospital admissions among the people in the two groups during the months following lockdown (see Table 3).

As can be seen in Table 3, the percentage of admissions at six months was lower and statistically significant in the intervention group (14% vs. 24.5%; χ^2^ = 4.577; *p* = 0.032). We also analyzed the effectiveness of the different subtypes of telematic intervention in reducing admissions. The percentage of admissions was significantly lower in those who received interventions by videoconference (4.2% vs. 22%; *p* = 0.025), and was lower, but with no statistical significance, in those who received interventions by telephone (14.4% vs. 23.3%; χ^2^ = 3.058; *p* = 0.080) and other telematic means (11.8% vs. 21.6%; χ^2^ = 1.986; *p* = 0.159).

Finally, the model obtained in the multilevel logistic regression analysis is shown in Table 4. First, the null model was created, which gave an ICC of 0.37. This variability justified the multilevel analysis. In the final model, it was observed that receiving intervention by videoconference was associated with less probability of hospital admission six months after lockdown (OR = 0.48; CI = 0.27–0.85). This lower probability was maintained when confounding variables were included, with a slight increase in the risk of admission associated with being a woman (OR = 1.42; CI = 1.01–2) and a reduction in risk associated with adherence to drug treatment (OR = 0.41; CI = 0.18–0.94) and face-to-face interventions (OR = 0.66; CI = 0.44–0.98).

## 4. Discussion

The study examined how nursing interventions changed during the COVID-19 pandemic and the extent to which TH was able to reduce relapse in this context.

For the general population, the COVID-19 pandemic has been a considerable source of stress and a contributory factor for increases in symptoms of depression and anxiety, especially in the most vulnerable subjects such as patients with SMI [21]. In this population, the role of nursing takes on special relevance since it is essential to ensure provision not only of the care that is already needed by SMI patients and their families, but also new care which may be required to deal with the stressors caused by the pandemic [22].

Our study showed that during the COVID-19 health crisis, mental health nursing interventions were forced to change, moving from a fundamentally face-to-face form of care to a predominance of telematic care, especially in the most severe period of lockdown at the beginning of the pandemic. Nursing interventions by telephone thus increased exponentially both during and after the period of lockdown. Video calls and other telematic platforms (blogs, social networks, and e-mail) were previously practically unused by nurses, but became important during lockdown. Their use decreased after the most severe period of lockdown, but has not returned to their pre-pandemic levels. The role played by nursing during the pandemic has been crucial to the implementation of new forms of patient care using the telephone, internet, video-call consultations, virtual support groups, etc. [1,23].

We believe that the greater use of the telephone was due to the fact that it is a medium known and used by all patients, while the other devices require more resources and knowledge to be used effectively. We also believe that the lack of a communicative environment able to ensure privacy and confidentiality may create a certain mistrust on the part of the patient and lead to a reluctance to use video calls. This type of care is not a new concept. Nurses with administrative tasks have been contacting their patients by telephone to communicate laboratory results, notify changes in medication, or give advice on care since 1970. In recent decades, these functions have been extended to include interventions such as providing training for health self-management, clarifying doubts about treatments, reminding patients of face-to-face appointments, and improving communication between health professionals [24,25]. For Locsin, the technological competence of nurses constitutes a type of nursing care in its own right, rather than being a modality through which they deliver care [26]. According to the World Health Organization, 70% of countries have opted for telematic assistance to make up for the lack of face-to-face care (“COVID-19 disrupting mental health services in most countries, WHO survey”) [26]. The most commonly used forms of TH are the telephone and image transfer via different platforms [24].

Our study has shown TH to be effective in reducing relapses in situations where face-to-face care is not possible, such as that generated by the pandemic and the lockdown. The effectiveness of these new media in nursing interventions has been demonstrated in another study, which reported how people with psychosis adhered more to treatment and reduced their number of relapses and visits to emergency services thanks to regular telephone consultations. In this case, telematic interventions helped patients cope with their illness, provided them with general health education and psycho-education regarding medication, and supported them in their decision making [27].

Regarding the different forms of telematic intervention, the reduction in relapses was greater in people who received interventions by videoconference. We believe that platforms which include video and/or audio facilitate a better perception of nonverbal language compared to other technological solutions. Together with the interpersonal relationship between patients and their doctors or nurses, this type of nonverbal communication is a key factor for understanding feelings and thoughts [25], making patients feel understood [18] and generating the perception that their needs are being met [26]. The study therefore also defends the importance of creating training programs for nurses on communication skills and empathy in remote consultations [28].

Even beyond the pandemic scenario, TH has its advantages, offering greater coverage of the population in rural or geographically dispersed areas [29], time savings, easy coordination between primary and community care, reduced waiting lists [26,30], acceptance and satisfaction among patients, and the perception by patients that their therapeutic relationships have not been negatively affected [15,17].

The research described in this paper provides an insight into how nursing interventions adapted to the circumstances during the first months of the pandemic. The use of MHDHs as active data collection centers resulted in a high level of data reliability, thanks principally to the availability of good registries and exhaustive patient information in these entities. The fact that this was a follow-up study allowed us to analyze changes over time in relation both to the inclusion of different channels of intervention and to the results in terms of relapses. Finally, the study’s multicenter design also facilitated greater generalization of the results obtained.

One of the main limitations of this work is that it was a retrospective study. As such, it was susceptible to certain biases, although an attempt was made to offset these by using objective variables based on medical records. Since it was an observational study, care should be taken not to establish causal relationships between variables. Further, the effectiveness of face-to-face interventions could not be compared to that of telematic interventions and it would have been interesting to provide more specific details of the interventions used.

With respect to future research, it would be important to use other outcome evaluation parameters more in line with the performance and recovery model. It would also be of interest to observe changes in intervention pathways and the specific interventions performed within each pathway over time. Regarding the effectiveness of telematic interventions outside the pandemic context, experimental studies will be required to test specific standardized interventions. It would also be important to assess the degree of satisfaction with telematic interventions among patients and nursing staff.

## 5. Conclusions

In conclusion, it can be said that although the COVID-19 pandemic has radically changed the ways in which mental health nursing interventions are carried out, with a reduction in face-to-face interventions, it also represents an opportunity to start using TH more extensively. In the pandemic context, the use of such telematic tools, and especially video calls, has been shown to have helped reduce later relapses.

Nevertheless, more research is still needed into COVID-19-related mental health nursing interventions and the effectiveness of TH outside the pandemic scenario.

## Figures and Tables

**Table 1 healthcare-10-00273-t001:** Description of sociodemographic data.

Variable	Category	Total *n* (%)	Patients Who Received Nursing Interventions	Patients Who Did Not Receive Nursing Interventions	*χ* ^2^	*p*
Diagnosis	Schizophrenia or other psychotic disorders	82 (30.4%)	32 (29.9%)	50 (30.7%)	6.502	0.165
	Bipolar disorder	28 (10.4%)	14 (13.1%)	14 (8.6%)		
	Personality disorder	75 (27.8%)	27 (25.2%)	48 (29.4%)		
	Major depressive disorder	26 (9.6%)	15 (14.0%)	11 (6.7%)		
	Other	59 (21.9%)	19 (17.8%)	40 (24.5%)		
Gender	Women	150 (55.6%)	60 (56.1%)	90 (55.2%)	0.019	0.889
	Men	120 (44.4%)	47 (43.9%)	73 (44.8%)		
Home composition	Complete family of origin	115 (42.6%)	50 (46.7%)	65 (39.9%)	11.537	0.009 *
	Own family home	78 (28.9%)	38 (35.5%)	40 (24.5%)		
	Single homeowner	46 (17%)	10 (9.3%)	36 (22.1%)		
	Other	31 (11.5%)	9 (8.4%)	22 (13.5%)		
Activity	Work/vocational/ occupational activity before the pandemic					
	Student	20 (7.4%)	6 (5.6%)	14 (8.6%)	19.637	0.001 *
	Temporary work disability	54 (20%)	25 (23.4%)	29 (17.8%)		
	Retired, pensioner	79 (29.3%)	32 (29.9%)	47 (28.8%)		
	Unemployed	71 (26.3%)	36 (33.6%)	35 (21.5%)		
	Working	45 (16.7%)	7 (6.5%)	38 (23.3%)		
	Volunteer/mutual aid agent	1 (0.4%)	1 (0.9%)	0 (0.0%)		
Level of Education	Primary	104 (38.5%)	45 (42.1%)	59 (36.2%)	5.785	0.216 *
	Secondary	112 (41.5%)	47 (43.9%)	65 (39.9%)		
	University	54 (20.0%)	15 (14.0%)	39 (23.9%)		

* *p* < 0.005.

**Table 2 healthcare-10-00273-t002:** Analysis of nursing interventions received pre-pandemic, during lockdown, and post-pandemic.

	% Patients Pre-Lockdown (January 16-March 15)	% Patients Lockdown (16 March–15 May)	% Patients Post-Lockdown (16 May–)	Before and during Lockdown		Before and after Lockdown		During and after Lockdown	
				χ^2^	*p* Value	χ^2^	*p* Value	χ^2^	*p* Value
Face-to-face nursing	59.6	16.3	60.4	104.3	0.001 *	0.02	0.888	113.2	0.001 *
Telephone nursing	4.1	33.3	6.7	75.11	0.001 *	30.420	0.001 *	20.338	0.001 *
Video call nursing	0	8.9	5.2		0.001 *		0.001 *		0.064
Nursing via other telematic interventions	4	12.6	9.3	29.257	0.001 *		0.001 *	2.370	0.124

* *p* < 0.001.

**Table 3 healthcare-10-00273-t003:** Analysis of hospital admissions in the six months following lockdown.

	Patients Who Received Nursing Interventions	Patients Who Did Not Receive Nursing Interventions		
Category			χ^2^	*p*
Telematic interventions	15 (14.0%)	40 (24.5%)	4.577	0.032
Video call interventions	1 (4.2%)	54 (22%)	5.718	0.025
Telephone interventions	13 (14.4%)	42 (23.3%)	3.058	0.080
Other telematic interventions	4 (11.8%)	51 (21.6%)	1.986	0.159

**Table 4 healthcare-10-00273-t004:** Multilevel logistic regression analysis to predict hospitalizations six months after lockdown.

	Model: Interventions + Sociodemographic Variables + Clinical Variables		
	β *(SE)*	*OR*	CI *95%*
Video call nursing	−0.735 (0.30)	0.48 *	0.27 to 0.85
Telephone nursing	−0.75 (0.24)	0.95	1.02 to 2.25
Nursing via other telematic interventions	−0.40 (0.28)	0.67	−0.45 to 1.16
Face-to-face nursing	−0.42 (0.2)	0.66 *	0.44 to 0.98
Female patient	0.35 (0,.18)	1.42 *	1.01 to 2
Age in years	0.01 (0.01)	1.01	0.99 to 1.01
Bipolar disorder	−0.57 (0.30)	0.56	0.31 to 1.03
Schizophrenia or other psychotic disorders	−0.09 (0.23)	0.91	0.58 to 1.43
Major depressive disorder	0.25 (0.32)	1.28	0.80 to 2.04
Others	0.24 (0.24)	1.28	0.68 to 2.39
Adherence	−0.88 (0.42)	0.41 *	0.18 to 0.94

*SE* = standard error; * *p* < 0.05; β = result of regression or beta equation; CI *95%* = confidence intervals.

## Data Availability

The data presented in this study are available on request from the corresponding author. The data are not publicly available because they are part of an ongoing project.

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
