# Peer review of "Nursing and Telemental Health during the COVID-19 Pandemic"

_healthcare, 2022, doi:10.3390/healthcare10020273_

Round 1

Reviewer 1 Report

I think study is well designed and results are clearly clarrified. However few minor comments

  1. I think abstract should have the form "introduction- materials and methods-results- conclusion".
  2. Abstract should give more details for the study and need to be more extended
  3. How authors were sure that the person on the phone was the patient and not someone who pretended the patient
  4. How authors ensured the confidentiality of the sessions
  5. How authors ensured that patients understood what they were asking during the telephone session

Reviewer 2 Report

This article is a study of a current issue, namely the impact of the pandemic on mental health, in this case on patients already suffering from mental illness. The study is of interest, although relatively simple, and offers new information on the topic. However, it appears globally inaccurate in form, which detracts from its quality.

The Introduction paragraph can be made more concise.

The order of the citations should be corrected (e.g. No 2 appears before No 1, and other minor errors are present in the text).

Line 57, the acronym 'TH' is given without previous reference (presumably to be included in line 50). The same applies to 'SMI' on line 92 and other acronyms across the text.

Line 65, report in numerical citation (Cowan et al., 2019; Smith et al., 2020).

Line 98 to line 106: I would consider moving the information to the Results section.

The Results section needs improvement. The text in places seems more like a statement of methods than results. The paragraph should be reorganized in a more systematic and reader-friendly way. A reference to Table 2 is missing in the text.

In the Discussion section the results are commented on too briefly and with too few references to other literature on the topic. This section should be implemented.

A clear definition of the paragraph 'Conclusions' is missing.

I recommend a revision of the work to make the exposition more systematic and the article more correct in its form, while also implementing the contents of the Discussion.

Author Response

This manuscript is a resubmission of an earlier submission. The following is a list of the peer review reports and author responses from that submission.